# Food groups, macronutrient intake and objective measures of total carotenoids and fatty acids in 16-to-24-year-olds following different plant-based diets compared to an omnivorous diet

Synne Groufh-Jacobsen[1]*, Christel Larsson[2], Isabelle Mulkerrins[2], Dagfinn Aune[3,4,5], Anine Christine Medin[1]

1 Department of Nutrition and Public Health, Faculty of Health and Sport Science, University of Agder, Kristiansand, Norway, 2 Department of Food and Nutrition, and Sport Science, Faculty of Education, University of Gothenburg, Sweden, 3 Department of Epidemiology and Biostatistics, School of Public Health, Imperial College London, London, United Kingdom, 4 Department of Nutrition, Oslo New University College, Oslo, Norway, 5 Department of Research, The Cancer Registry of Norway, Norwegian Institute of Public Health, Oslo, Norway

* synne.groufh.jacobsen@uia.no

**Data Availability Statement:** Data from this study contains potentially identifiable participant

## Abstract

### Background

Knowledge about the diet quality among youth who follow different types of plant-based diets is essential to understand whether support is required to ensure a well-planned diet that meets their nutritional needs. This study aimed to investigate how food groups, macronutrient intake, and objective blood measures varied between Norwegian youth following different plant-based diets compared to omnivorous diet.

### Methods

Cross-sectional design, with healthy 16-to-24-year-olds (n = 165) recruited from the Agder area in Norway, following a vegan, lacto-ovo-vegetarian, pescatarian, flexitarian or omnivore diet. Participants completed an electronic questionnaire, a dietary screener, 24-hour dietary recalls and provided dried blood samples for analysis of carotenoids and fatty acids.

### Results

Vegans reported the highest mean intake (g/d, g/MJ) of vegetables, legumes, nuts and seeds and substitutes to dairy and meat (compared to all, p<**0.001**), fruit and berries (compared to omnivores, p = **0.004** and pescatarians, p = **0.007**), and vegetable oil (compared to omnivores, p<**0.001**, pescatarians, p = **0.003** and flexitarians, p = **0.004**) and vegetable products (compared to omnivores, p = **0.007**). No difference was found between groups in mean intake (g/d, g/MJ) of any of the confectionary foods or sweet pastries, beverages (sugar-sweetened, non-sugary, alcoholic), or salted snacks, neither in g/MJ of convenience

information that cannot be shared publicly due to ethical and legal restrictions. These restrictions are imposed by the Regional Committees for Medical and Health Research Ethics (REK) in Norway, in accordance with Norwegian privacy laws, including the Personal Data Act and GDPR. Public sharing of the data, even in de-identified form, could risk compromising participant confidentiality due to the specific nature of the data including a small sample size with a minority group (plant-based dietary practice). Requests for access to the anonymized data set can be directed to the Data Access Committee at the University of Agder, which manages data sharing in compliance with these ethical regulations. Request for data access can be made to the privacy protection officer at the University of Agder by contacting trond. hauso@uia.no. All requests will be reviewed by the principal investigator in the project to ensure compliance with the original ethical approvals and privacy protections.

**Funding:** The author(s) received no specific funding for this work.

**Competing interests:** The authors have declared that no competing interests exist.

foods. The energy percentage (E%) of protein, carbohydrates and total fat were within the Nordic Nutrition Recommendations 2023 across groups. However, all groups, except vegans, exceeded the E% for saturated fatty acids. All groups exceeded recommendations for added and free sugar. Furthermore, all groups consumed <25g/d of dietary fibre, except vegans and pescatarians. For omega-3, lacto-ovo-vegetarians had intakes below recommendations. Blood marker of total carotenoids did not differ between groups, neither did the reported mean intake (g/MJ) of carotenoid-rich foods. Vegans showed the lowest blood level of palmitic acid compared to all **(p<0.001)**, but highest level of linoleic acid (compared to flexitarians, p = **0.022**, and omnivores, p<**0.001**). The lowest blood levels of eicosapentaenoic acid and docosahexaenoic acid were found in vegans and lacto-ovo-vegetarians.

## Conclusions

Our findings suggest that all groups had risk of dietary shortcomings. However, vegans consumed the most favorable diet. All groups should increase their consumption of vegetables, fruits and berries, and reduce their total sugar intake.

## Introduction

Globally, noncommunicable diseases (NCDs) are the leading cause of premature mortality, in which unhealthy diets is a driver towards several risk factors of developing NCDs in western countries [1, 2]. An optimal dietary pattern has been described in the Nordic Nutrition Recommendations 2023 (NNR2023) as a diet that is adequate in nutrients, reduces the risk of NCDs, and at the same time reduces the impact of our dietary choice on climate [3]. NNR2023 states that the average Nordic diet today exceeds the planetary boundaries and emphasizes that a diet characterized by high intakes of vegetables, fruits, whole grains, fish, low-fat dairy and legumes, and low intakes of red and processed meat, sugar-sweetened beverages, sugary foods, and refined grains promotes health [3]. A review from the USDA Nutrition Evidence Systematic Review team supports the described dietary pattern, with strong evidence for reduction of all-cause mortality [4]. Additionally, a shift towards a more plant-based diet is emphasized in NNR2023 as a strategy to reduce the environmental impact of our diet [3]. Likewise, the World Health Organization also recommends increased consumption of plant-based foods to secure a more sustainable diet [5]. Additionally, a reduction in animal-sourced foods has been suggested as more sustainable food consumption compared to higher meat consumption [6–11].

Today, nutrition societies around the world supports that plant-based diets can be nutritionally adequate throughout all stages of the life course as long as the diet is well-planned [12]. In the literature, a well-planned plant-based diet is described as a diet that meets an individual's dietary needs, and is suggested to consist of various vegetables, fruits, whole grains, legumes, as well as nuts and seeds [5]. However, there are still ongoing controversies about the risk of following plant-based diets in childhood and adolescence [13–17], especially vegan diets [18]. Bakaloudi and co-workers have pointed out that the nutrient adequacy of a diet depends on the types and quantities of food groups that are included in the diet [18]. A systematic review published in 2017 investigated the risk of vegetarian diets in childhood to adolescence, and the authors concluded that there is no consensus on the health benefits and risk of following plant-based diets in this period of life [16]. According to a new systematic review published in 2023 that examined the dietary intake and status of children and adolescents in

the age range of 2–18 years, there are nutritional challenges in both plant-based dietary groups and omnivores [17]. This confirms previous evidence that nutrient adequacy depends on the food groups consumed in the diet regardless of dietary practice [18].

A previous Norwegian national survey from 2021 showed an increasing interest in plant-based diets, especially among younger people, particularly females [19]. However, data on the food habits of youth who follows different types of plant-based diets in Nordic countries are limited as the previous study that examined the dietary intake including details on food group intake was conducted over a decade ago [20]. Since then, the availability of plant-based alternatives has changed to large extent, and hence the food habits of youth who follow plant-based diets could also have changed. The youth life-phase is a period of transition from dependence in childhood to adulthood's independence. Therefore, youth need to be sufficiently equipped with the necessary competencies to meet their dietary requirements without the help of their parental guardians [21, 22]. Updated knowledge on food habits among young people who follow different types of plant-based diets in Nordic countries is needed.

To our knowledge, the VeggiSkills-Norway project, in which this study is part of, is the first providing data on diet quality among Norwegian youth following different types of plant-based and omnivore diets. Therefore, these data will provide insight on whether dietary support is needed for youth to secure a well-planned diet. This study aimed to investigate how food groups, macronutrient intake, and objective blood measurements of carotenoids and fatty acids varied between youth following different types of plant-based diets compared to an omnivorous diet.

## Methods

### Study design and study population

This study is a part of a larger cross-sectional research project called VeggiSkills-Norway [23]. Data collection was carried out between September 2021 and March 2022. Healthy 16–to-24-year-olds with different dietary practices (vegan, lacto-ovo-vegetarian, pescatarian, flexitarian, and omnivore) from the Agder area in Norway were invited to participate in the study. A priori, the sample size was determined for the VeggiSkills-Norway project based on a previously conducted study among young Swedish vegans and omnivores that detected dietary intake differences with 30 participants in each dietary group [20]. Thus, the aim was to recruit 240 participants in the VeggiSkills-Norway project, with 60 participants within each dietary group (60 vegans, 60 lacto-ovo-vegetarians, 60 pescatarians, 60 omnivores) to be able to perform analysis split by sex. In the planning phase, we decided to include flexitarians as well, as this was reported as the most prevalent plant-based dietary practice at national level, and dietary intake and status data for flexitarians are also lacking among Norwegian youth. Our final sample size was determined by the recruitment feasibility since the study took place during the COVID-19 pandemic.

To be included in the study, participants had to 'be able to read and understand Norwegian', 'be 16-to-24 years old', 'have no acute or chronic illness', 'currently not pregnant / lactating or have children', 'had to have followed their respective diet for a minimum of 6 months prior to participation', 'no current plan to alter their diet', and 'had to be able to physically attend the measurements at the University of Agder, Kristiansand, Norway'. Participants were recruited using convenience and snowball sampling methods, and the recruitment process and study design have been fully described elsewhere [24, 25]. Participants who were eligible for participation, and who filled out a study consent, were then asked to complete an electronic questionnaire and a dietary screener [MinMatMåned 1.1], during a site visit at the University of Agder. Subsequently, during the same site visit, all participants performed an initial 24-hour

dietary recall (details described in the 24-hour dietary recall section), which was interviewer-administered. Thereafter, (during the same site visit), participants were asked to provide a non-fasted dried blood sample (DBS) and a urinary sample for analysis of nutritional status biomarkers. Body composition measurements were also performed during the same site visit. After the study visit, participants were asked to complete three self-administered non-consecutive 24-hour dietary recalls in a web-based program, Myfood24, over an eight-week period.

## Classification into dietary practice

Participants were classified into the different dietary practices according to their self-reported inclusion of animal source foods the previous six months in the electronic questionnaire, in which participants were asked to report their consumption frequency of the following food groups, egg/egg products, milk and dairy products, fish and seafood, meat/poultry/meat products [26]. Based on the reported frequency, participants were classified into vegan (no consumption of animal-source foods), lacto-ovo-vegetarian (includes dairy or egg/egg products to varying degree), pescatarian (includes dairy or egg/egg products and seafood to varying degree), flexitarian (consuming any type of meat less than two servings per week [27]) and omnivore (reported two or more servings of any type of meat per week).

The reported information in the electronic questionnaire was subsequently cross-checked with the reported consumption of animal source foods in a dietary screener [MinMatMåned 1.1 [28]] completed by the participants, which assessed habitual intake of 33 food groups the previous six months, as previously described and the intake data from the dietary screener is published elsewhere [24, 25].

The categorizing into the different dietary practices was performed prior to evaluating the 24-hour dietary recalls. Hence, six participants reported inconsistency with their dietary practice after evaluating the 24-hour dietary recalls, in which one vegan reported consumption of eggs and one lacto-ovo-vegetarian reported consumption of fish products. Furthermore, one vegan, one lacto-ovo-vegetarian, and one pescatarian reported consumption of processed meat products, and one pescatarian reported consumption of red and white meat in the 24-hour dietary recalls. These participants were not re-classified, as the aforementioned methods were used for classification into the different dietary practices and used in previous analysis in the VeggiSkills-project [24, 25].

In the final sample, 19 participants were classified as vegans, 20 as lacto-ovo-vegetarians, 30 as pescatarians, 25 as flexitarians and 71 as omnivores.

## 24-hour dietary recalls

The validated image-assistant dietary assessment tool Measure Your Food in One Day (Myfood24) [29] was used to assess their dietary intake presented in this study. Participants were asked to complete four 24-hour dietary recalls on random non-consecutive days over an eight-week period using Myfood24. Myfood24 provides a calculation of the energy, and nutrients of the foods and beverages reported consumed. The Norwegian version of Myfood24 which was used is connected to the Norwegian Food Composition database [30].

The first 24-hour dietary recall was carried out at the study site and was conducted through an interview with the same researcher using the multiple-pass method, where all reported foods, beverages, and supplements were entered directly into Myfood24. The first recall lasted between 25–50 minutes. The multiple-pass method is based on an approach to improve the accuracy of the dietary reporting [31]. Participants were first told to give a brief summary of all foods and beverages consumed the previous day. The mealtimes (breakfast, lunch, dinner, in-between meals) were collected, and in cases where the participants had difficulty remembering,

the researcher helped the participants to clarify time and eating location. After the foods and beverages consumed the previous day had been reported, the researcher went through the intake and collected a more detailed description, and probes were used to minimize the risk that the participants had forgotten anything. Subsequent to the study visit, participants received unannounced messages to complete the remaining three 24-hour dietary recalls with roughly one-week intervals. A maximum of two reminders were given for each dietary recall. The dietary recalls were distributed on different days to include both weekdays and weekends.

During the first 24-hour dietary recall, participants were given brief instructions on how to perform the self-administered recalls with the dietary assessment tool Myfood24. If a specific food item was not available in the Myfood24 database, participants were told to choose a similar food/beverage alternative. In addition, it was also possible to report foods and beverages that were not available in the Myfood24 database as an open text option. Foods and beverages reported in the open text field were manually processed. For example, if a person reported consuming a vegan product mainly consisting of coconut milk and vegetable oil, raw ingredients from coconut milk and vegetable oil were entered to obtain a similar energy and macronutrient profile of the product. The self-administered 24-hour dietary recalls were performed without supervision, thus the duration for completing these recalls is unknown.

## Categorization of food items into food groups

All foods and beverages reported consumed were given a unique food code from the Myfood24 database. The food codes were categorized into 33 food groups presented in **S1 Table**, in which 29 of the food groups are presented in the manuscript. The food items were categorized into food subcategories already available in Myfood24, however, some changes were made to the existing food subcategories (for detail see **S2 Table**). The categorization into the carotenoid-rich food groups (**S3 Table**) to further investigate participants fruit and vegetable intakes, was completed according to a categorization used in a previous validation study among Norwegian children and adolescents [32].

## Blood sampling and biochemical analysis

In this study, participants were asked to provide non-fasted capillary blood using the dried blood sampling method (DBS). The DBS cards were allowed to air dry in a room without direct light exposure before being placed in an aluminum bag with a desiccant pack at -20 degrees Celsius (°C) until analysis (maximum drying time 12 hours). The DBS were analyzed at VITAS™ Analytical Services, Norway, and in this present study, carotenoids and fatty acids are presented. No established international cut-offs exist for carotenoids and fatty acids measured in DBS. Therefore, the blood levels in this study are presented descriptively.

Carotenoids were analyzed using HPLC-UV, in which five punches from the DBS were eluted in 70 μL of water and then precipitated with 300μL of isopropanol (including internal standards for the different compound groups). The resulting mixture was thoroughly mixed and centrifuged. The supernatant obtained was injected into an HPLC system with a HP 1100 liquid chromatograph (Agilent Technologies, Palo Alta, CA, USA) and a HP1260 Infinity diode array detector set at 453 nm. Carotenoids were separated on a 4.6 mm x 150 mm reversed phase C-30 column, maintained at a temperature of 50°C. A one-point calibration curve was made from the analysis of plasma calibrators with known concentrations of carotenoids. Carotenoids analyzed in the DBS were lutein, zeaxanthin, β-cryptoxanthin, ß-carotene, α-carotene, lycopene and total carotenoids (the sum of lutein, zeaxanthin, β-cryptoxanthin, ß-carotene, α-carotene, lycopene).

Fatty acids were analyzed using GC-FID, in which two punches from the DBS were methylated with sodium methylate. Following methylation, fatty acid methyl esters (FAME) were extracted using hexane. After thorough mixing and centrifugation, a 3 μl of the aliquot was injected into the GC-FID, and the GC-FID analysis was performed with an HP 7890A gas chromatography system (Agilent Technologies, Palo Alto, CA, USA). The fatty acid methyl esters were separated on a TR-FAME column from Thermo Scientific 30m x 0.25mm x 0.25μm column. Fatty acids analyzed in DBS were palmitic acid, stearic acid, oleic acid, linoleic acid, arachidonic acid, γ-linoleic acid, gamma-linolenic acid, DHA, EPA, docosapentaenoic acid (DPA), alpha-linolenic acid (ALA).

## Anthropometric measurements

All participants had their height measured while standing barefoot, using a portable stadiometer. Body weight was measured in a nonfasted state, barefoot, and in light clothing, using a Tanita MC780 bioelectric impedance analyzer (Tokyo, Japan).

## Statistics

SPSS Statistics version 29 (IBM Corp., Armonk, NY, United States) was used for statistical analysis. Excel ® (Microsoft 365) was used for data processing of food codes. The normality of the data was checked using visual inspection of the histogram and Q-Q plots. Nonparametric data are presented as median and percentiles (25th,75th) and parametric data as mean ± standard deviation (SD). To test for differences between the groups using continuous nonparametric variables, Mann-Whitney U test was used if comparing two groups and Kruskal-Walli's test if comparing more than three groups with pairwise comparison. To test between the groups using parametric variables, One-way ANOVA with Bonferroni Post Hoc test for correction of multiple comparisons was used. Cross-tabulation using the Fisher exact test was run to test for the difference between groups using categorical variables. The two-sided p-value <0.05 was used as the significance level for all tests.

The energy percentage (E%) was calculated for protein, carbohydrates, added and free sugar, total fat, saturated fatty acids (SFA), polyunsaturated fatty acids (PUFA), omega-3 and omega-6 fatty acids, and monounsaturated fatty acids (MUFA), and then compared with the NNR2023 [3]. Most of the food group intakes were non-normally distributed within the dietary groups, and zero intakes were reported for several food groups. Therefore, sensitivity analysis was performed evaluating both mean and median values and using both parametric and nonparametric tests to test for difference between groups. Mean ± SD values are presented in the manuscript to make comparison with existing dietary data possible. Median (25th,75th) food group intake values g/d and g/MJ are presented in **S4 and S5 Tables** for transparency as the data was non-normally distributed. Animal source foods are included in the tables for descriptive purposes, the overall p-value for difference between groups is not displayed due to no consumption assumed among vegans. Differences in the consumption of animal source foods from the 24-hour dietary between consuming groups are indicated in the manuscript. For descriptive purpose, food group intake was stratified by sex shown in g/d in **S6 Table** and in g/MJ in **S7 Table**.

## Ethics

Before the start of the study, written informed consent was obtained from all participants, and the study was conducted according to the guidelines laid down in the Declaration of Helsinki. The term 'youth' is used to describe the study population (16-to-24-year) in VeggiSkills-Norway. The United Nation (UN) defines youth as those who are between the age of 15–24 years

[21]. The minimum age of 16 years was used in our study for practical considerations, so that the participants could provide their own study consent without the approval from their parental guardians. The research ethics committee does not require study consent from parental guardians for subjects between the ages of 16–17 years in Norway. All procedures involving research with the participants were approved by the Norwegian Regional Committees for Medical and Health Research Ethics (REK/217742) and the Norwegian Centre for Research Data (NSD/168890).

## Results

### Participant characteristics

Participant characteristics are presented in Table 1. Out of 165 participants, 76% were females, and the mean age was 21±2 with lowest age among omnivores (20±2 years), and highest in pescatarians (22±2 years) (p = **0.010**). Body mass index (BMI) did not differ significantly between the dietary groups. At the group level, a mean number of three 24-hour dietary recalls were completed, and the number completed between the dietary groups did not differ (p = 0.19).

### Food groups

Categorization of the food items into food groups is presented in S1–S3 Tables. The food groups are presented in Table 2 as mean absolute intakes (g/d) and as energy-adjusted intake (g/MJ) in Table 3. Sensitivity analysis for median intakes (g/d and g/MJ) are presented in S4 and S5 Tables. For descriptive purpose, food group intake stratified by sex and food group intake stratified by sex within the different dietary practices (g/d and g/MJ) are presented in S6 and S7 Tables. The mean and median absolute intake (g/d) and energy-adjusted intakes (g/MJ) of the carotenoid-rich food groups are presented in S8 and S9 Tables.

**Table 1. Participant characteristics of 16–to–24-year-olds in the VeggiSkills-Norway study with different diet (n = 165).**

| | All[†] | | Vegans[†] | | Lacto-ovo-vegetarians[†] | | Pescatarians[†] | | Flexitarians[†] | | Omnivores[†] | | P-value |
|---|---|---|---|---|---|---|---|---|---|---|---|---|---|
| | n | % | n | % | n | % | n | % | n | % | n | % | |
| Female[‡] | 125 | 76 | 13 | 68 | 19 | 95 | 26 | 87 | 22 | 88 | 45 | 63 | **0.005** |
| Male[‡] | 40 | 24 | 6 | 32 | 1 | 5 | 4 | 13 | 3 | 12 | 26 | 37 | |
| 24-hour dietary recall day 1[¶] | 165 | 100 | 19 | 100 | 20 | 100 | 30 | 100 | 25 | 100 | 71 | 100 | |
| 24-hour dietary recall day 2[¶] | 130 | 79 | 16 | 85 | 24 | 83 | 16 | 53 | 19 | 76 | 38 | 53 | |
| 24-hour dietary recall day 3[¶] | 99 | 60 | 14 | 74 | 12 | 60 | 16 | 53 | 19 | 76 | 38 | 54 | |
| 24-hour dietary recall day 4[¶] | 72 | 44 | 10 | 53 | 8 | 40 | 11 | 37 | 14 | 56 | 29 | 41 | |
| | Mean | SD | Mean | SD | Mean | SD | Mean | SD | Mean | SD | Mean | SD | |
| Number of 24-hour dietary recalls[§] | 3 | 2 | 4 | 1 | 3 | 1 | 3 | 1 | 3 | 1 | 3 | 1 | 0.19 |
| Age, years[§] | 21 | 2 | 22 | 2 | 21 | 2 | 22* | 2 | 22 | 2 | 20[‖] | 2 | **0.010** |
| Body mass index, kg/m²[§] | 23 | 4 | 22 | 3 | 23 | 5 | 24 | 3 | 22 | 4 | 24 | 3 | 0.32 |

Abbreviations: SD = Standard deviation

[†]Percentage presented within each dietary group

[‡]Test for the difference (categorical variables) using cross tabulation with Fisher-Exact Test

[¶]Differences between the dietary groups in the number of completed 24-hour dietary recalls are reported in the variable 'number of dietary recalls'

[§]Test for the difference (continuous variables) using one-way ANOVA with Bonferroni Post Hoc test correction for multiple comparisons, unlike superscript indicate differences ([‖],*); Statistically significant values between the dietary groups <0.05 are given in bold (two-sided). For more detailed participant characteristics see previous publications from the VeggiSkills-Norway project [24, 25].

**Table 2. Mean food group intake g/d by repeated 24-hour dietary recalls in Norwegian 16-to-24-year-olds with different dietary practice.**

| Food group intake, g/d | All | | Vegans | | Lacto-ovo-vegetarians | | Pescatarians | | Flexitarians | | Omnivores | | P-value |
|---|---|---|---|---|---|---|---|---|---|---|---|---|---|
| (Absolute intake) | n = 165 | | n = 19 | | n = 20 | | n = 30 | | n = 25 | | n = 71 | | |
| | Mean | SD | Mean | SD | Mean | SD | Mean | SD | Mean | SD | Mean | SD | |
| **Plant-sourced foods** | | | | | | | | | | | | | |
| Whole grain products, g/d‡ | 87 | 69 | 99 | 89 | 76 | 54 | 95 | 59 | 96 | 62 | 82 | 73 | 0.68 |
| Refined grain products, g/d‡ | 72 | 67 | 105 | 98 | 53 | 44 | 78 | 56 | 58 | 58 | 72 | 68 | 0.11 |
| Vegetables (all types), g/d‡ | 102 | 82 | 182* | 117 | 87† | 65 | 118† | 74 | 105† | 71 | 78† | 68 | **<0.001** |
| Fruit and berries (not including juice/smoothie), g/d‡ | 151 | 150 | 267* | 240 | 134† | 90 | 121† | 119 | 169 | 136 | 131† | 138 | **0.005** |
| Legumes, g/d‡ | 19 | 39 | 58* | 75 | 33*,† | 37 | 25†,§ | 41 | 13†,§ | 22 | 6§ | 17 | **<0.001** |
| Nuts and seeds, g/d‡ | 8 | 18 | 35* | 34 | 4† | 8 | 5† | 12 | 6† | 12 | 3† | 10 | **<0.001** |
| Vegetable oil, g/d‡ | 2 | 4 | 5* | 7 | 2 | 3 | 1† | 2 | 2† | 3 | 1† | 3 | **<0.001** |
| Potatoes (including sweet potatoes) g/d‡,∣ | 19 | 33 | 16 | 27 | 9 | 18 | 20 | 33 | 21 | 30 | 22 | 39 | 0.61 |
| Vegetable products, g/d‡ | 14 | 27 | 33* | 51 | 9 | 13 | 16 | 24 | 14 | 29 | 10† | 17 | **0.015** |
| Fruit and berry products, g/d‡ | 4 | 15 | 12 | 24 | 6 | 25 | 2 | 5 | 1 | 3 | 3 | 12 | 0.07 |
| Dairy product substitutes, g/d‡ | 25 | 94 | 123* | 232 | 18† | 47 | 12† | 24 | 38† | 88 | 3† | 12 | **<0.001** |
| Meat substitutes and vegetarian food products, g/d‡,¶ | 19 | 46 | 80* | 93 | 22† | 30 | 19† | 31 | 13† | 36 | 4† | 18 | **<0.001** |
| Vegetarian dishes, g/d‡ | 34 | 160 | 20 | 51 | 48 | 90 | 100 | 358 | 13 | 25 | 14 | 37 | 0.14 |
| **Animal-sourced foods** | | | | | | | | | | | | | |
| Milk and dairy products (including cheese), g/d‡ | 157 | 179 | 0 | 1 | 125 | 150 | 151 | 137 | 143 | 103 | 216 | 217 | – |
| Eggs, all types, g/d‡ | 25 | 40 | 2 | 7 | 30 | 35 | 26 | 44 | 30 | 43 | 28 | 42 | – |
| Red meat, all types, g/d‡ | 20 | 43 | 0 | 0 | 0 | 1 | 0 | 1 | 17 | 28 | 41 | 57 | – |
| White meat, all types, g/d‡ | 11 | 31 | 0 | 0 | 0 | 0 | 1 | 5 | 11 | 25 | 22 | 42 | – |
| Lean, fatty fish and shellfish, g/d‡ | 29 | 51 | 0 | 0 | 0 | 2 | 46 | 68 | 45 | 56 | 31 | 50 | – |
| Fish products, g/d‡ᵃ | 19 | 43 | 0 | 0 | 0 | 2 | 30 | 60 | 24 | 44 | 22 | 44 | – |
| Butter/margarine, g/d‡ | 5 | 11 | 2 | 6 | 4 | 5 | 6 | 8 | 8 | 25 | 4 | 6 | – |
| **Sugary, salted and convenience foods** | | | | | | | | | | | | | |
| Dessert, cake, and sweets, g/d‡ | 55 | 68 | 23 | 31 | 53 | 47 | 67 | 54 | 47 | 36 | 63 | 89 | 0.17 |
| Sweetened bread spread, g/d‡ | 5 | 10 | 6 | 13 | 4 | 6 | 6 | 9 | 5 | 8 | 4 | 11 | 0.88 |
| Sweetened cereal, g/d‡ | 8 | 17 | 7 | 17 | 7 | 12 | 4 | 12 | 9 | 19 | 9 | 20 | 0.67 |
| Salted snacks, g/d‡ | 9 | 20 | 14 | 26 | 8 | 13 | 7 | 11 | 9 | 23 | 9 | 22 | 0.82 |
| Convenience foods, g/d‡,ᵃ | 75 | 126 | 11 | 28 | 81 | 155 | 87 | 136 | 35 | 62 | 99 | 138 | **0.033** |
| **Beverages** | | | | | | | | | | | | | |
| Alcoholic beverages, g/d‡ | 35 | 110 | 41 | 133 | 22 | 68 | 44 | 152 | 55 | 154 | 26 | 68 | 0.77 |
| Non-sugary beverages, g/d‡ | 130 | 270 | 109 | 191 | 79 | 180 | 121 | 181 | 77 | 209 | 172 | 347 | 0.48 |
| Juice and smoothie, g/d‡ | 53 | 97 | 62 | 87 | 57 | 95 | 48 | 87 | 39 | 49 | 55 | 117 | 0.93 |
| Sugar-sweetened beverages, g/d‡ | 74 | 155 | 14 | 43 | 67 | 129 | 53 | 112 | 44 | 59 | 112 | 205 | 0.07 |

‡Test for difference using one-way ANOVA with Bonferroni Post Hoc test with correction for multiple comparisons, unlike superscript indicate differences (*,†,§)
(overall p-value not displayed for milk and dairy products, red meat, white meat, processed meat products, lean, fatty fish and shellfish, fish products, no significant difference in post hoc test between consuming groups); Statistically significant values between the dietary groups <0.05 are given in bold (two-sided)

∣Not including processed (fried) potatoes (included in the convenience food category)

¶ In addition to meat substitutes the food items 'hummus', 'sesame paste, tahini', 'vegetable pâté, Tartex'are included. For a description of food items included in the food groups, see **S1 Table**

ᵃ Non-significant in post-hoc test adjusted for multiple comparison.

**Table 3. Mean food group intake g/MJ by repeated 24-hour dietary recalls in 16-to-24-year-olds with different dietary practice.**

| Food group intake, g/MJ | All | | Vegans | | Lacto-ovo-vegetarians | | Pescatarians | | Flexitarians | | Omnivores | | P-value |
|---|---|---|---|---|---|---|---|---|---|---|---|---|---|
| (Energy-adjusted) | n = 165 | | n = 19 | | n = 20 | | n = 30 | | n = 25 | | n = 71 | | |
| | Mean | SD | Mean | SD | Mean | SD | Mean | SD | Mean | SD | Mean | SD | |
| **Plant-sourced foods** | | | | | | | | | | | | | |
| Whole grain products, g/MJ[‡] | 12 | 9 | 12 | 8 | 13 | 10 | 13 | 9 | 15 | 10 | 11 | 10 | 0.45 |
| Refined grain products, g/MJ[‡] | 10 | 9 | 14 | 15 | 10 | 11 | 10 | 7 | 8 | 7 | 9 | 8 | 0.20 |
| Vegetables (all types), g/MJ[‡] | 14 | 11 | 24* | 13 | 15[†] | 11 | 15[†] | 9 | 16[†] | 9 | 11[†] | 9 | <**0.001** |
| Fruit and berries (not including juice/smoothie), g/MJ[‡] | 21 | 20 | 34* | 28 | 23 | 16 | 16[†] | 16 | 24 | 17 | 18[†] | 19 | **0.015** |
| Legumes, g/MJ[‡] | 3 | 5 | 8* | 8 | 6*,[†] | 7 | 3[†,§] | 4 | 2[†,§] | 4 | 1[§] | 2 | <**0.001** |
| Nuts and seeds, g/MJ[‡] | 1 | 2 | 4* | 3 | 1[†] | 1 | 1[†] | 1 | 1[†] | 1 | 0[†] | 1 | <**0.001** |
| Vegetable oil, g/MJ[‡] | 0 | 1 | 1* | 1 | 0 | 1 | 0[†] | 0 | 0[†] | 0 | 0[†] | 0 | <**0.001** |
| Potatoes (including sweet potatoes), g/MJ[‡\|] | 3 | 5 | 2 | 4 | 1 | 3 | 3 | 4 | 3 | 5 | 3 | 7 | 0.71 |
| Vegetable products, g/MJ[‡] | 2 | 4 | 4* | 7 | 2 | 2 | 2 | 3 | 2 | 5 | 1[†] | 2 | **0.039** |
| Fruit and berry products, g/MJ[‡] | 1 | 3 | 2 | 3 | 2 | 8 | 0 | 1 | 0 | 1 | 0 | 2 | 0.19 |
| Dairy product substitutes, g/MJ[‡] | 4 | 13 | 18* | 33 | 3[†] | 7 | 2[†] | 3 | 5[†] | 10 | 0[†] | 2 | <**0.001** |
| Meat substitutes and vegetarian food products, g/MJ[‡¶] | 3 | 5 | 10 | 9 | 4 | 5 | 2 | 4 | 1 | 3 | 1 | 3 | <**0.001** |
| Vegetarian dishes, g/MJ[‡] | 5 | 19 | 3 | 9 | 9 | 15 | 12 | 41 | 2 | 5 | 2 | 6 | 0.15 |
| **Animal-sourced foods** | | | | | | | | | | | | | |
| Milk and dairy products (including cheese), g/MJ[‡] | 22 | 24 | 0 | 0 | 24 | 35 | 20 | 19 | 22 | 16 | 29 | 24 | – |
| Eggs (all types), g/MJ[‡] | 4 | 6 | 0 | 1 | 6 | 9 | 4 | 5 | 5 | 7 | 4 | 6 | – |
| Red meat (all types), g/MJ[‡] | 3 | 5 | 0 | 0 | 0 | 0 | 0 | 0 | 3 | 4 | 5 | 6 | – |
| White meat (all types), g/MJ[‡] | 2 | 4 | 0 | 0 | 0 | 0 | 0 | 1 | 1 | 2 | 3 | 6 | – |
| Lean, fatty fish and shellfish, g/MJ[‡] | 4 | 7 | 0 | 0 | 0 | 0 | 6 | 7 | 6 | 9 | 4 | 7 | – |
| Fish products, g/MJ[‡] | 2 | 5 | 0 | 0 | 0 | 0 | 3 | 6 | 4 | 7 | 3 | 6 | – |
| Butter/margarine, g/MJ[‡] | 1 | 1 | 0 | 1 | 1 | 1 | 1 | 1 | 1 | 1 | 1 | 1 | – |
| **Sugary, salted and convenience foods** | | | | | | | | | | | | | |
| Dessert, cake, and sweets, g/MJ[‡] | 7 | 7 | 3 | 4 | 9 | 7 | 8 | 6 | 7 | 4 | 7 | 9 | 0.11 |
| Sweetened bread spread, g/MJ[‡] | 1 | 1 | 1 | 2 | 1 | 1 | 1 | 1 | 1 | 1 | 1 | 2 | 0.99 |
| Sweetened cereal, g/MJ[‡] | 1 | 2 | 1 | 2 | 1 | 2 | 1 | 1 | 1 | 2 | 1 | 2 | 0.63 |
| Salted snacks, g/MJ[‡] | 1 | 2 | 1 | 2 | 1 | 2 | 1 | 1 | 1 | 3 | 1 | 3 | 0.97 |
| Convenience foods, g/MJ[‡] | 11 | 18 | 1 | 4 | 13 | 25 | 12 | 18 | 6 | 12 | 13 | 19 | 0.08 |
| **Beverages** | | | | | | | | | | | | | |
| Alcoholic beverages, g/MJ[‡] | 5 | 13 | 5 | 17 | 3 | 10 | 4 | 14 | 7 | 18 | 4 | 10 | 0.79 |
| Non-sugary beverages, g/MJ[‡] | 19 | 41 | 17 | 29 | 13 | 28 | 17 | 26 | 12 | 31 | 24 | 53 | 0.71 |
| Juice and smoothie, g/MJ[‡] | 7 | 12 | 9 | 13 | 9 | 16 | 6 | 11 | 6 | 8 | 7 | 13 | 0.87 |
| Sugar-sweetened beverages, g/MJ[‡] | 10 | 20 | 2 | 4 | 11 | 21 | 6 | 13 | 6 | 9 | 15 | 26 | 0.05 |

[‡]Test for the difference using one-way ANOVA with Bonferroni Post Hoc test correction for multiple comparisons, unlike superscript indicate differences (*,[†],[§]) (overall p-value not displayed for milk and dairy products, red meat, white meat, processed meat products, lean, fatty fish and shellfish, fish products, no significant difference in post hoc test between consuming groups)

[\|]Not including processed/ pre-prepared (fried) potatoes (included in the convenience food category)

[¶] In addition to meat substitutes the food items 'hummus', 'sesame paste, tahini', 'vegetable pâté, Tartex' are included; Statistically significant values between the dietary groups <0.05 are given in bold (two-sided). For food items included in the food groups see **S1 Table**.

**Plant-based sourced foods.** The mean intake of plant-based foods which varied significantly between the dietary groups for both absolute intake (g/d) and energy-adjusted intake (g/MJ) were vegetables (p<**0.001**), fruit and berries (p = **0.005**), legumes (p<**0.001**), nuts and seeds (p<**0.001**), vegetable oil (p<**0.001**), dairy product substitutes (p<**0.001**), meat

substitutes and vegetarian food products (p<**0.001**) and vegetable products (p = **0.015**). No significant difference was found between the dietary groups in mean intake (both in g/d and g/MJ) of whole grain products, refined grain products, potatoes, fruit and berry products and vegetarian dishes.

Vegans reported the highest mean intake compared to all the other dietary groups in g/d for the following food groups, vegetables at 182±117 g/d (lowest among omnivores at 78±68 g/d, p<**0.001**), legumes at 58±75 g/d (lowest among omnivores at 6±17 g/d, p<**0.001**), nuts and seeds at 35±34 g/d (lowest among omnivores at 3±10 g/d, p<**0.001**), dairy product substitutes at 123±232 g/d (lowest among omnivores at 3±12 g/d, p<**0.001**), and meat substitutes and vegetarian food products at 80±93 g/d (lowest among omnivores at 4±18 g/d, p<**0.001**). Vegans also reported the highest mean intake of fruit and berries at 267±240 g/d compared to all groups except flexitarians (lowest among pescatarians at 121±119 g/d, p = **0.007**). Furthermore, vegans also reported the highest mean intake of vegetable oil at 5±7 g/d, compared to all groups except lacto-ovo-vegetarians (lowest among omnivores at 1±3 g/d). Although some of the reported intake of vegetable oil includes raw ingredients from vegan substitutes (see method for details, 24-hour dietary recall section). Additionally, vegans reported the highest mean intake of vegetable products at 33±1 g/d, compared to omnivores at 10±17 g/d (p = **0.007**). Lacto-ovo-vegetarians also reported a higher mean intake of legumes at 33±37 g/d, compared to omnivores at 6±17 g/d (p = **0.035**). After energy-adjustment (g/MJ) the food groups still differed significantly, except fruit and berries (g/MJ), which did not remain significantly different between vegans and lacto-ovo-vegetarians (p = 0.92).

Sensitivity analysis evaluating the median intake values and nonparametric test for difference between groups showed nearly the same results, except for fruit and berries (g/d, p = 0.07) and vegetable products (g/d, p = 0.18), with no difference between groups. Furthermore, the difference in intake of legumes (g/d) between vegans and pescatarians did not persist (p = 0.20). And for vegetable oil, median intake in g/d among vegans only differed significantly from omnivores (p = **0.025**).

For reported absolute intake (g/d) of the carotenoid-rich food groups, a significant difference was found for total carotenoid rich foods (p = **0.025**). In which, vegans reported higher intake of total carotenoid-rich foods in g/d compared to omnivores (p = **0.009**), but this difference did not remain statistically significant after energy-adjustment (g/MJ) (**S8 Table**). Sensitivity analysis evaluating median intakes (**S9 Table**) showed that absolute intake (g/d) of ß-cryptoxanthin-rich foods (p = **0.042**, no difference between groups in post hoc test) and lutein+zeaxanthin-rich foods (p = **0.031**, varied between vegans and omnivores p = **0.046**) varied significantly between the dietary groups. When evaluating the energy-adjusted median intakes (g/MJ), intakes ß-cryptoxanthin-rich foods and lutein+zeaxanthin-rich foods still varied significantly.

**Animal-sourced foods.** For the animal-sourced foods, lowest and highest mean intake ±SD (g/d) is presented for consuming groups (not including vegans for any of the animal-sourced food groups, and not lacto-ovo-vegetarians for seafood or red and white meat, and not pescatarians for red and white meat).

No significant difference was found between any of the consuming groups for milk and dairy, eggs, lean fish, fatty fish and shellfish, fish products, red meat, or white meat (both in g/d and g/MJ). The reported mean intake of milk and dairy (including cheese) was lowest among lacto-ovo-vegetarians at 125±150 g/d and highest in omnivores at 216±217 g/d. Mean egg intake was lowest among pescatarians at 26±44 g/d and highest in lacto-ovo-vegetarians at 30±35 g/d. Furthermore, mean intake of lean fish, fatty fish and shellfish was lowest among omnivores at 31±50 g/d, and highest in pescatarians at 46±68 g/d. The mean intake of fish products was lowest among omnivores at 22±44 g/d and highest in pescatarians at 30±60 g/d.

The mean intake of red meat (all types, including processed meat) was lower among flexitarians than among omnivores, 17±28 g/d, respectively, 41±57 g/d (no significant difference between the two consuming groups). White meat (all types, including processed meat) was lower among flexitarians at 11±25 g/d, and highest in omnivores at 22±42 g/d. After energy-adjustment (g/MJ), there was still no difference in any of the animal-sourced food groups between the consuming groups (vegans not included), similar results were found in the sensitivity analysis (median values and non-parametric tests).

For mixed dishes that includes meat (see description in **S1 Table**), flexitarians reported a mean intake at 13±33 g/d (2±5 g/MJ) and omnivores a mean intake at 35±61 g/d (5±9 g/MJ), with no statistical difference between the two consuming groups in the post hoc test (g/d, p = 0.40, g/MJ, p = 0.64).

**Sugary foods, salted snacks, convenience foods and beverages.** No significant difference was found between the dietary groups in reported intake (both in g/d and g/MJ) for dessert, cake and sweets, sweetened bread spread, sweetened cereal, salted snack, alcoholic beverages, non-sugary beverages, juice and smoothies or sugar-sweetened beverages. A difference was found between the dietary groups in intake of convenience foods in g/d, although this was not significant when energy-adjusted (g/MJ). Sensitivity analysis evaluating median values and non-parametric test for difference showed similar results, except, vegans had significantly lower median intake (g/MJ) of convenience foods compared to omnivores (p = **0.016**). Furthermore, sensitivity analysis showed that vegans had significantly lower median intake (g/MJ) of dessert, cake and sweets compared to pescatarians (p = **0.011**).

## Energy and macronutrient intake

The energy and macronutrient intake (from foods and supplements) are presented in **Table 4**. No difference was found between the dietary groups in the reported mean energy intake in MJ/day. Sensitivity analysis showed differences in median energy intake between lacto-ovo-vegetarians (6 MJ/day) compared to omnivores (7 MJ/day, p = 0.011), and between lacto-ovo-vegetarians compared to pescatarians (8MJ/day, 0.008).

**Protein.** The E% from protein was lowest among vegans (13E%), followed by lacto-ovo-vegetarians (14E%) and pescatarians (15E%), which differed significantly from omnivores (17E%) who had the highest E% (p<**0.001**). All groups (including flexitarians, 16E%) were within the recommendation of 10-20E%.

**Total carbohydrates, added/free sugar, and dietary fibre.** The E% from carbohydrates was lowest among flexitarians (43E%) and highest in lacto-ovo-vegetarians (50E%) (p = **0.035**). All groups, except for flexitarians, were within the recommended 45-60E%. For added and free sugar combined, all groups exceeded 10E%, and no differences were found between the groups. For dietary fibre, only vegans and pescatarians achieved the recommended daily intake of 25 g/d for females, and all groups reported significantly lower intakes compared to vegans (p<**0.001**). Vegans was the only dietary group reporting intakes that met the daily dietary fiber recommendation for men as well (at least 35 g/d), at 39±22 g/d.

**Fat.** The E% of total fat did not differ between the groups and all groups were within the recommended 25-40E%. Vegans reported the lowest E% of SFA (8E%) compared to all the other groups (p<**0.001**). All groups, except vegans, exceeded the recommendation of <10E% of SFA in the diet. Vegans reported the highest E% from PUFA at 9E%, compared to all the other dietary groups (p<**0.001**), and all dietary groups had mean intakes within the recommendation of 5-10E% of PUFA. Vegans had significantly higher E% of omega-6 fatty acids at 8E%, compared to all the other dietary groups (p<**0.001**), although all groups were within the omega-6 fatty acid recommendation of 3E% There was no difference in E% between the

**Table 4. Energy and macronutrient intake by repeated 24-hour recalls in Norwegian 16-to-24-year-olds with different dietary practice.**

| | NNR2023 | All | | Vegans | | Lacto-ovo-vegetarians | | Pescatarians | | Flexitarians | | Omnivores | | p-value |
|---|---|---|---|---|---|---|---|---|---|---|---|---|---|---|
| | | n = 165 | | n = 19 | | n = 20 | | n = 30 | | n = 25 | | n = 71 | | |
| | | Mean | SD | Mean | SD | Mean | SD | Mean | SD | Mean | SD | Mean | SD | |
| Energy intake, MJ [‖‡¶] | | 7 | 3 | 8 | 4 | 6 | 1 | 8 | 2 | 7 | 3 | 8 | 3 | 0.05 |
| Energy intake, Kcal[‡] | | 1768 | 645 | 1905 | 947 | 1405 | 332 | 1852 | 431 | 1659 | 811 | 1836 | 598 | |
| Protein, E%[‡,] | 10-20E% | 16 | 4 | 13[†] | 4 | 14[†] | 3 | 15[†] | 3 | 16 | 3 | 17[*] | 4 | <**0.001** |
| Carbohydrates, E%[‡] | 45-60E% | 46 | 8 | 48 | 8 | 50 | 5 | 46 | 7 | 44 | 7 | 45 | 8 | 0.49 |
| Added sugar and free sugar, E%[‡] | <10E% | 18 | 7 | 19 | 8 | 21 | 7 | 15 | 6 | 17 | 4 | 18 | 8 | 0.09 |
| Added sugar only, E%[‡] | | 8 | 12 | 5 | 8 | 7 | 6 | 6 | 5 | 11 | 27 | 7 | 7 | 0.45 |
| Dietary fiber (g/d)[‡‖] | | 24 | 12 | 39[*] | 22 | 21[†] | 9 | 26[†] | 9 | 23[†] | 12 | 21[†] | 8 | <**0.001** |
| Total fat, E%[‡] | 25-40E% | 35 | 7 | 34 | 6 | 32 | 5 | 36 | 7 | 35 | 7 | 35 | 8 | 0.41 |
| SFA, E%[‡] | <10E% | 12 | 4 | 8[*] | 2 | 12[†] | 3 | 12[†] | 4 | 13[†] | 4 | 12[†] | 3 | <**0.001** |
| PUFA, E%[‡] | 5-10E% | 6 | 2 | 9[*] | 2 | 5[†] | 2 | 7[†] | 2 | 6[†] | 2 | 6[†] | 2 | <**0.001** |
| Omega-3[‡] | 1E% | 1.2 | 0.8 | 1.5 | 0.8 | 0.8 | 0.6 | 1.3 | 0.7 | 1.2 | 0.7 | 1.2 | 0.9 | 0.05 |
| Omega-6[‡] | 3E% | 5 | 2 | 8[*] | 2 | 5[†] | 2 | 5[†] | 2 | 5[†] | 2 | 5[†] | 2 | <**0.001** |
| MUFA, E%[‡] | 10-20E% | 13 | 4 | 13 | 5 | 12 | 2 | 14 | 4 | 13 | 4 | 13 | 4 | 0.31 |
| Trans fat, E%[‡] | As low as possible | 0.2 | 0.2 | 0.1[*] | 0.1 | 0.3[†] | 0.2 | 0.3[†] | 0.1 | 0.3[†] | 0.1 | 0.3[†] | 0.2 | <**0.001** |

Abbreviations: NNR2023 = Nordic Nutrition Recommendations 2023; E% = Energy percentage; SFA = saturated fatty acids; PUFA = polyunsaturated fatty acids; MUFA = monounsaturated fatty acids

[‖] Recommended energy intake for adult females 9MJ/d and males 11.3 MJ/d, and dietary fibre for females at least 25 g/d and 35 g/d for males according to Nordic Nutrition recommendation 2023 [3]

[¶] Sensitivity analysis evaluating median values showed differences in median energy intake between lacto-ovo-vegetarians (6 MJ/day) compared to omnivores (7MJ/day, p = 0.011), and between lacto-ovo-vegetarians compared to pescatarians (8MJ/day, 0.008)

[‡] Test for the difference using one-way ANOVA with Bonferroni Post Hoc test with correction for multiple comparisons, unlike superscript indicate differences ([†,*]); Statistically significant values between the dietary groups <0.05 are given in bold (two-sided). Decimals are given for trans fat and omega-3 for meaningful values.

groups for omega-3 fatty acids, but lacto-ovo-vegetarians were the only group who reported intake (0.8E%) below the recommendation of 1E%. Furthermore, there was no difference in E % between groups for MUFA, and all groups were within the recommendation of 10-20E% for MUFA.

## Objective blood markers

**Carotenoids.** Blood markers of carotenoids are presented in **Table 5**. No difference were found between the dietary groups for non-fasting whole blood levels of total carotenoids in μmol/L (displaying the sum of lutein, zeaxanthin, β-cryptoxanthin, ß-carotene, α-carotene, lycopene) (p = 0.34). Furthermore, no difference were observed between the dietary groups for the carotenoids separately, except for lutein (p = **0.021**), in which omnivores had significantly lower levels compared to flexitarians in the post hoc test (0.16±0.07 vs. 0.22±0.08 μmol/L, p = **0.024**).

**Fatty acids.** Blood markers of fatty acids are presented in **Table 5**. The fatty acid profile measured in whole blood (weight% of fatty acid ester (FAME) in whole blood) varied between the groups for palmitic acid (SFA), linoleic acid (PUFA, omega-6) (p<**0.001**), arachidonic acid (PUFA, omega-6) (p = **0.003**), EPA (PUFA, omega-3) (p<**0.001**), and DHA (PUFA, omega-3) (p<**0.001**), DPA (PUFA, omega-3) (p = **0.007**). No difference was found between the dietary groups for stearic acid (SFA), oleic acid (MUFA), alpha-linolenic acid (PUFA, omega-3), γ-linoleic acid (PUFA, omega-6) or gamma-linolenic acid (PUFA, omega-6).

**Table 5. Carotenoids and fatty acids measured in whole blood in Norwegian 16-to-24-year-olds with different dietary practice.**

| Markers | All | | | Vegans | | | Lacto-ovo-vegetarians | | | Pescatarians | | | Flexitarians | | | Omnivores | | | P-value |
|---|---|---|---|---|---|---|---|---|---|---|---|---|---|---|---|---|---|---|---|
| | Mean | SD | n | Mean | SD | n | Mean | SD | n | Mean | SD | n | Mean | SD | n | Mean | SD | n | |
| **Carotenoid, whole blood in µmol/L** | | | | | | | | | | | | | | | | | | | |
| Total carotenoids‡ | 1.34 | 0.59 | 146 | 1.36 | 0.74 | 17 | 1.20 | 0.39 | 19 | 1.38 | 0.45 | 28 | 1.56 | 0.73 | 23 | 1.29 | 0.59 | 59 | 0.34 |
| Lutein‡ | 0.17 | 0.07 | 135 | 0.19 | 0.07 | 17 | 0.16 | 0.05 | 18 | 0.18 | 0.07 | 28 | 0.22* | 0.08 | 20 | 0.16† | 0.07 | 52 | **0.021** |
| Zeaxanthin‡ | 0.05 | 0.02 | 144 | 0.05 | 0.03 | 17 | 0.05 | 0.02 | 18 | 0.04 | 0.02 | 28 | 0.05 | 0.02 | 20 | 0.05 | 0.02 | 58 | 0.29 |
| β-cryptoxanthin‡ | 0.13 | 0.09 | 146 | 0.14 | 0.09 | 17 | 0.14 | 0.15 | 19 | 0.12 | 0.06 | 28 | 0.14 | 0.08 | 23 | 0.12 | 0.11 | 59 | 0.76 |
| α-carotene‡ | 0.11 | 0.26 | 146 | 0.12 | 0.12 | 17 | 0.08 | 0.04 | 19 | 0.09 | 0.06 | 28 | 0.15 | 0.12 | 23 | 0.11 | 0.14 | 59 | 0.35 |
| β-carotene‡ | 0.37 | 0.26 | 146 | 0.35 | 0.32 | 17 | 0.31 | 0.21 | 19 | 0.37 | 0.15 | 28 | 0.48 | 0.35 | 23 | 0.35 | 0.24 | 59 | 0.23 |
| Lycopene‡ | 0.53 | 0.24 | 145 | 0.49 | 0.32 | 17 | 0.47 | 0.26 | 19 | 0.57 | 0.26 | 28 | 0.55 | 0.23 | 23 | 0.53 | 0.22 | 58 | 0.71 |
| **Fatty acids, whole blood in weight% of FAME** | | | | | | | | | | | | | | | | | | | |
| Palmitic acid (SFA)‡ | 21.10 | 1.67 | 163 | 19.22* | 2.02 | 19 | 20.95† | 1.24 | 19 | 21.45† | 1.58 | 30 | 21.16† | 1.49 | 25 | 21.49† | 1.44 | 70 | **<0.001** |
| Stearic acid (SFA)‡ | 11.78 | 1.06 | 163 | 11.96 | 1.05 | 19 | 11.97 | 0.87 | 19 | 11.63 | 1.09 | 30 | 11.82 | 1.32 | 25 | 11.73 | 1.01 | 70 | 0.75 |
| Oleic acid (MUFA)‡ | 21.69 | 2.46 | 163 | 22.75 | 2.72 | 19 | 22.44 | 3.47 | 19 | 21.33 | 1.92 | 30 | 21.35 | 2.75 | 25 | 21.49 | 2.43 | 70 | 0.14 |
| Linoleic acid (PUFA, Omega-6)‡ | 25.10 | 2.55 | 163 | 27.31* | 3.22 | 19 | 25.93*¶ | 2.28 | 19 | 25.43 | 2.22 | 30 | 25.06¶ | 2.60 | 25 | 24.15† | 2.08 | 70 | **<0.001** |
| Arachidonic acid (PUFA, Omega-6)‡ | 11.44 | 1.75 | 163 | 11.38 | 1.76 | 19 | 11.06 | 1.34 | 19 | 10.53† | 1.75 | 30 | 11.40 | 1.72 | 25 | 11.96* | 1.70 | 70 | **0.003** |
| γ-linoleic acid (PUFA, Omega-6)‡ | 0.25 | 0.11 | 163 | 0.23 | 0.08 | 19 | 0.22 | 0.09 | 19 | 0.26 | 0.12 | 30 | 0.25 | 0.12 | 25 | 0.25 | 0.12 | 70 | 0.76 |
| Gamma-linolenic acid (PUFA, Omega-6)‡ | 1.79 | 0.43 | 163 | 1.99 | 0.37 | 19 | 1.87 | 0.48 | 19 | 1.86 | 0.46 | 30 | 1.68 | 0.48 | 25 | 1.72 | 0.37 | 70 | 0.05 |
| Eicosapentaenoic (EPA) (PUFA, Omega-3)‡ | 1.14 | 0.77 | 163 | 0.61* | 0.21 | 19 | 0.75* | 0.37 | 19 | 1.41† | 1.02 | 30 | 1.31† | 0.94 | 25 | 1.21† | 0.65 | 70 | **<0.001** |
| Docosapentaenoic acid (DPA) (PUFA, Omega-3)‡ | 1.46 | 0.31 | 163 | 1.31* | 0.24 | 19 | 1.33 | 0.27 | 19 | 1.44 | 0.32 | 30 | 1.47 | 0.33 | 25 | 1.55† | 0.31 | 70 | **0.007** |
| Docosahexaenoic acid (DHA) (PUFA, Omega-3)‡ | 3.59 | 1.04 | 163 | 2.44* | 0.93 | 19 | 2.85* | 0.73 | 19 | 4.02† | 0.83 | 30 | 3.87† | 1.00 | 25 | 3.83† | 0.93 | 70 | **<0.001** |
| Alpha-linolenic acid (ALA) (PUFA, Omega-3)‡ | 0.65 | 0.29 | 163 | 0.79 | 0.49 | 19 | 0.65 | 0.21 | 19 | 0.65 | 0.31 | 30 | 0.62 | 0.25 | 25 | 0.63 | 0.23 | 70 | 0.28 |

Abbreviations: FAME = Fatty acids methyl esters; SFA = saturated fatty acids; MUFA = monounsaturated fatty acids, PUFA = polyunsaturated fatty acids ‡Test for the difference using One-way ANOVA with Bonferroni Post Hoc test with correction for multiple comparisons, unlike superscript indicate differences (*,†,¶); Statistically significant values between the dietary groups <0.05 are given in bold (two-sided).

Vegans showed significantly lower blood levels of palmitic acid compared to all the other dietary groups (compared to lacto-ovo-vegetarians p = **0.007,** and compared to pescatarians, flexitarians and omnivores p<**0.001**). Vegans showed significantly higher blood levels of linoleic acid compared to flexitarians (p = **0.022**) and omnivores (p<**0.001**). Omnivores showed a significantly higher blood level of arachidonic acid compared to pescatarians (p = **0.001**). Furthermore, EPA levels was significantly lower in vegans compared to pescatarians (p = **0.003**), flexitarians (p = **0.021**) and omnivores (p = **0.020**), and EPA levels was also significantly lower among lacto-ovo-vegetarians compared to pescatarians (p = **0.024**). DHA levels were significantly lower in vegans and lacto-ovo-vegetarians compared to all the other dietary groups (pescatarians, flexitarians and omnivores, p<**0.001**, except between lacto-ovo-vegetarians and flexitarians p = **0.003**). For DPA levels, vegans showed significantly lower levels compared to omnivores (p = **0.025**).

## Discussion

### Summary of main findings

In this study, the reported intake of food groups and macronutrients varied between the youth with different dietary practice, vegan, lacto-ovo-vegetarian, pescatarian, flexitarian and omnivore. Statistically significant differences in mean intake of plant-sourced foods, both in absolute intake (g/d) and energy-adjusted intake (g/MJ) were found between the dietary groups for the following food groups, vegetables, fruit and berries, legumes, nuts and seeds, vegetable oil, dairy product substitutes and meat substitutes and vegetarian food products, and also vegetable products. In which, the highest reported mean absolute intake (g/d) of these food groups were reported among vegans, and lowest among omnivores (except for fruit and berries, in which the lowest intake in g/d was reported by pescatarians). For animal-sourced foods, no difference was found in mean absolute intake (g/d) or energy-adjusted intake (g/MJ) between the consuming groups (not including vegans). Furthermore, no difference was found between the dietary groups for mean absolute intake (g/d) of dessert, cake and sweets, sweetened bread spread, sweetened cereal, salted snacks, alcoholic beverages, non-sugary beverages, juice and smoothies or sugar-sweetened beverages, with similar findings when evaluating the energy-adjusted intakes (g/MJ). Intake of convenience foods differed significantly in g/d between groups, but this did not remain after energy-adjustment (g/MJ). Additionally, most of the macronutrient recommendations from NNR2023 were met [3], except for energy (all groups below), SFA (all groups exceeded, except vegans), omega-3 (lacto-ovo-vegetarians below), dietary fibre (only met by vegans and pescatarians), and all groups exceeded the recommendation for added and free sugar. Furthermore, no significant differences were found between the dietary groups in blood level of carotenoids, supportive of the reported consumption of carotenoid-rich foods, in which no significant difference was found between groups in g/MJ. For blood levels of fatty acids, levels of DHA and EPA were significantly lower in vegans and lacto-ovo-vegetarians, compared to all the other dietary groups. Furthermore, vegans and lacto-ovo-vegetarians showed higher blood levels of linoleic acid (omega-6 fatty acid). Vegans displayed the lowest blood level of palmitic acid (SFA) in alignment with the dietary data showing the lowest mean consumption of SFA among vegans.

### Food group intake among youth following plant-based diets

Despite numerous of systematic reviews pointing out that the nutrient adequacy of plant-based diets depends on the type and quantity of food groups consumed, and currently, there is very limited knowledge on the food group intake among young people who adhere to plant-based diets [16–18, 33]. In the most recent systematic review among children and adolescents

comparing plant-based diets to omnivores [17], six out of 33 studies had subjects in the age range of 16–18 years [34–39], in which only three of them examined food group intake [34, 35, 39], and the remaining three examined iron status. The limited number of studies available among youth who follow different types of plant-based diets highlights the need for more studies.

One of the studies that examined food group intake was published in 2002 by Larsson et al, among Swedish vegans and omnivores in the age range 16–20 years (vegans n = 30, omnivores n = 30) [35]. Similar to our findings, the study found the highest consumption of vegetables and legumes among vegans compared to omnivores. Additionally, no difference was reported between the Swedish vegans and omnivores for consumption of pizza, pie and pastries, chips and popcorn, soft drinks or alcoholic beverages, in coherence with our finding, in which no significant difference were found in sweetened bread spread, sweetened cereal, salted snacks, alcoholic beverages, non-sugary beverages, juice and smoothies, sugar-sweetened beverages or convenience foods (convenience food varied significantly in g/d, but not in g/MJ). Contrasting to our findings, significantly lower consumption of cakes, cookies, candy and chocolate was reported among the Swedish vegans compared to the omnivores [35]. Although the lowest consumption of these foods was also reported among the vegans in our study, the intake did not differ significantly between the dietary groups. It also needs to be noted that the Swedish study was conducted two decades ago. Since then, many new plant-based substitutes have become available on the food market in the past decades, therefore, our findings are not directly comparable with this study. Although this study is the only study available on food group consumption in a Nordic youth population with a plant-based diet compared to omnivores.

One of the other studies, the VeChi youth study conducted in Germany, was published in 2021, including vegans (n = 110), lacto-ovo-vegetarians (n = 145), and omnivores (n = 135) in the age range 6–18 years [34] (number of participants in the age range 16–18 years not reported). The VeChi youth showed similar findings to ours, in which the vegans displayed the highest median consumption in g/MJ of vegetables, legumes, nuts, and plant-based substitutes to dairy. However, in contrast to their findings, we did not find significant differences in median intake in g/MJ of whole grain products or any of the sugary foods (data presented in the supplemental material). Furthermore, participants in the VeChi youth study [34] had a mean age of 13±4 years with all age groups pooled into one sample, compared to 21±2 years in our study, the age difference makes comparison to our study challenging.

A third study was conducted in 2019 among females, aged 15–18 years, who were self-identified vegetarians (n = 38) and omnivores (n = 216) in New Zealand [39], but the food group intakes in that study are presented in percentage among consumers and not in g/d or g/MJ. The study reported lower consumption of animal-source foods and higher consumption of plant-based substitutes among the self-identified vegetarians when compared to the omnivores, which is an expected finding. Furthermore, they also reported that 13 out of 38 self-identified vegetarians reported inconsistencies with their dietary practice, reporting consumption of meat, poultry and fish or seafood, similarly to the six participants in our study that reported small intakes of animal-source foods, inconsistent with their dietary practice (see method/dietary categorization). Although full dietary compliance was not found in our study, the vast majority reported consistent diet categories by the two dietary assessment methods (dietary screener and 24-hour dietary recalls). Since few participants in our study reported inconsistency with their dietary practice, it is considered to not have a substantial impact on our study findings.

## Replacement of animal-sourced foods

The mean consumption of substitutes to dairy products and meat, as well as vegetarian food products was found to be highest among the participating vegans. The reported mean energy-adjusted intake (g/MJ) of meat substitutes among vegans was equivalent to the reported mean consumption (g/MJ) of red and white meat (all types) among the omnivores. But, for substitutes to dairy products, the mean consumption in g/MJ among vegans was not equivalent to the mean consumption in g/MJ of milk and dairy products among the omnivores. This suggests that the plant-based substitutes did not fully replace milk and dairy product, coherent with findings in the VeChi youth study [34]. Furthermore, in our study, we found the highest mean energy-adjusted intake (g/MJ) of legumes, nuts and seeds, and vegetables, fruit and berries, vegetable products and vegetable oil among the vegans. While omnivores had the lowest consumption of these food groups (except for fruit and berries, in which pescatarians had the lowest consumption). This suggests that vegans also replace the animal-sourced foods with the aforementioned food groups to some extent.

Lacto-ovo-vegetarians in our study reported nearly similar mean energy-adjusted intake of milk and dairy products (g/MJ) as omnivores (no significant difference between these groups), this finding is in contrast to the VeChi youth study, in which lacto-ovo-vegetarians had significantly lower median consumption of dairy products compared to the omnivores in g/MJ [34]. Although when evaluating the median consumption in g/MJ and not mean intake in g/MJ in our study, the omnivores in our study reported two-fold the consumption of dairy products (albeit still not significant). Lacto-ovo-vegetarians in our study also reported a significantly higher mean consumption in g/MJ of legumes compared to omnivores, suggesting that meat was replaced with legumes to some extent. Furthermore, pescatarians did not report an intake of meat substitutes equivalent to the reported red and white meat consumption among the omnivores, however, the higher consumption of legumes among pescatarians compared to omnivores (although not significant) suggests that legumes was used to replace meat to some extent.

## Food group intake in the general population

In our study, notably higher mean absolute intakes (g/d) were reported among the omnivores for fruit and berries, sweetened cereal, eggs and non-sugary beverages compared to the findings of the National Report among Norwegian adolescents, conducted in 2015 (13-year-olds, Ungkost 3) (n = 687, 52% females) [40]. Additionally, lower mean absolute intakes (g/d) among the participating omnivores in our study was found for vegetables, potatoes, juice and smoothie, red meat, milk and dairy and for sugar-sweetened beverages [40]. Furthermore, nearly similar mean absolute intakes (g/d) were found for nuts and seeds and salted snacks, white meat, and for dessert, cakes, and sweets among the omnivores in our study and the Norwegian adolescents in Ungkost 3.

Furthermore, a new report from the Norwegian Institute of Public Health investigated dietary habits of adolescents (14 year-olds) prior to, during, and after the COVID-19 pandemic (n = 38 368, 52% females) [41]. They found an increase in the sugar intake among those with already high sugar intakes during the COVID-19 pandemic, with more frequent increase among adolescent females. This is in contrast to our study findings, as we found no significant difference for any of the sugary foods or beverages in mean g/d or g/MJ between females and males on group level. The VeggiSkills-Norway study was conducted during the end of the COVID-19 pandemic and most of the participants are females, thus, there is a possibility that our findings of sugar intake above the NNR2023 recommendation (added and free sugar exceeding the recommendation in all dietary groups) might be due to the timing of the data collection and the results need to be interpreted with consideration to this.

Compared to the absolute intakes (g/d) in nationally representative dietary data on adults aged 18–70 years (n = 1787, 52% females), lower mean intake (g/d) was found for vegetables, fruit and berries, potatoes, meat (red and white, all types), milk and dairy products, but higher intakes of juice and smoothies, eggs, and snacks among the omnivores in our study [42]. Importantly, it should be noted that the Norkost 3 data was collected between 2010 and 2011, with 16% in the age range 18–29.

Additionally, between 2016–2019, it was found that Norwegian female adolescents (14–17 years) tended to report higher consumption of vegetables, fruit and berries and sweets compared to males (n = 25 996, 49% females) [43]. In our study, when stratified by sex (total sample) (data presented in supplemental material), the young females reported higher mean absolute intakes (g/d) of whole grain products, fruit and berries, dairy product substitutes, meat substitutes and vegetarian food products, vegetarian dishes compared to the male participants. Furthermore, mean absolute intake (g/d) of refined grain products, legumes, red meat (all types) and convenience food were significantly higher among male participants in our study compared to the females. Thus, our findings need to be interpreted with this information kept in mind, as the majority of the study population are females. Notably there are also fewer males in the plant-based dietary groups compared to the omnivore group, which may impact their reported food group intake. Food group intake is presented for descriptive purposes stratified by sex within the different dietary groups in the S6 and S7 Tables, due to power issue it is not possible to test for differences stratified by sex within the dietary groups.

## Macronutrient intake

The macronutrient profile within the different dietary practices was in line with the recommendations by NNR2023, except for energy intake, SFA, dietary fibre, omega-3 and added/free sugar. Two previous systematic reviews on dietary intake in children and adolescents (age range 2–18 years [17] and 0–18 years [44]) comparing plant-based diets to omnivores, corroborate our findings of lower E% from protein and SFA in vegans compared to omnivores, and the finding of higher E% from PUFA and dietary fibre (g/d) among the vegans compared to the omnivores. Similar results were found in a systematic review of 141 studies from 2022 on nutrient intake and status in adults (>18 years) eating plant-based and omnivorous diets [45], supporting our finding of lower protein and higher intakes of fiber and PUFA among the plant-based diet groups (especially vegans) compared to omnivores.

## Objective blood markers of total carotenoids and fatty acids

People who adhere to plant-based diets have previously been found to have a higher serum carotenoid concentration, especially of ß-carotene [46]. In our study no significant differences were detected between the dietary practices for whole blood carotenoids separately (except for lutein) or for the total carotenoid level (displaying the sum of the carotenoids). Previous studies evaluating biomarkers of fruits and vegetables in children and adolescents have reported various findings on correlations between vegetable and fruit intakes with blood levels of carotenoids, with mainly no correlations or weak correlations reported [47]. To our knowledge, there are few studies on blood carotenoid levels in a youth population comparing plant-based diets to omnivores, but a previous study with Finnish adults [48] reported the highest blood levels of ß-carotenoids among omnivores. In our study, vegans reported two times the intake (g/d) of vegetables compared to the omnivores, although no significant difference after energy-adjustment (g/MJ). It needs to be noted that the carotenoids are a concentration biomarker not directly reflecting the reported intake in the 24-hour dietary recalls, and the blood

measurements of carotenoids presented in this study were intended to be used as an objective indicator of the habitual consumption of carotenoid rich-fruit and vegetables. Since the carotenoids are measured in red blood cells using DBS it provides an indication of the habitual consumption, as the average life span of red blood cells is 120 days. Although, several factors could influence the bioavailability of carotenoids measured in whole blood, including co-ingestion of fats, as fat consumption can facilitate the uptake of carotenoids, although we did not find differences between the dietary groups for total fat intakes in E%. Moreover, vegans reported the highest fibre consumption, and a high fibre consumption can also inhibit the uptake of carotenoids. Furthermore, the bioavailability of carotenoids is also affected by heat and mechanical treatment of carotenoid-rich foods, which was not taken into consideration in this study [46]. Smoking and BMI can also influence the bioavailability of carotenoids, but there were no significant differences in these variables between the dietary groups. Lack of significant differences in for example BMI could potentially also be a power issue due to lower number of participants in the plant-based dietary groups (background variables previously presented [24]).

Furthermore, we found that the levels of palmitic acid (SFA) (typical food sources: butter, cheese, beef, pork, tropical oils) and arachidonic acid (SFA) (typical food sources: poultry, animal organs, meat, fish, seafood, and eggs), linoleic acid (omega-6 fatty acid) (typical food sources: vegetable oils, nuts seeds), and omega-3 fatty acids (EPA, DPA, DHA) (typical food sources: fatty fish/ fish oil supplements) varied between the dietary groups. A previous study found that DHA and EPA levels were significantly altered when measured in plasma, but not when measured in red blood cells [49]. In our study, the fatty acids were measured in red blood cells and can therefore possibly reflect their habitual consumption. In our study, vegans and lacto-ovo-vegetarians had significantly lower levels of EPA and DHA compared to the other dietary groups, as expected due to the few dietary sources in these respective dietary groups (typical food sources: fatty fish/ fish oil supplements). Limited data on the level of fatty acids are available in young people who follow different types of plant-based diets. A systematic review in adults comparing plant-based diets with omnivores reported lower levels of DHA and EPA in vegans and lacto-ovo-vegetarians compared to omnivores, while higher levels of ALA were reported in vegans and lacto-ovo-vegetarians [45]. This is similar to our findings, except for ALA levels that did not differ between dietary groups. ALA (typical food sources: vegetable oils, walnuts, flaxseeds) can be metabolized into DHA and EPA, although at a lower conversion rate, which can be impaired by high intakes of omega-6 fatty acids [49]. The vegans in this present study reported significantly higher E% of omega-6 fatty acids compared to the other dietary groups and reported more than two times the recommended intake of omega-6 fatty acids. Furthermore, we found that vegans had the lowest levels of palmitic acids, supported by the self-reported dietary intake data. The finding of the lowest level of SFA intake and palmitic acids level in blood among vegans are coherent with previous findings in adults both for blood markers [50, 51] and for the dietary data [52].

### Strengths and limitations of the study

The main strength of this study is the detailed dietary data using repeated 24-hour dietary recalls together with objective blood markers. Misreporting is however a common issue when using self-reported dietary data. To reduce the impact of day-to-day variation when using the 24-hour dietary recall method, repeated non-consecutive recalls were collected [53]. Furthermore, in our study a previously validated tool with image assisted portion sizes was used to improve the accuracy of the dietary reporting [28]. Another strength of this study is that the categorization into the dietary groups is not based on self-identified dietary practice, instead

participants were categorized based on their reported intake of animal-sourced foods in the previous six months which was also cross-checked with their reported consumption frequency of foods and beverages in a previously validated dietary screener [MinMatMåned 1.1].

The most critical limitation in our study is the small sample size, especially in the plant-based dietary groups, which might increase the risk of type II error. The recruitment method is another limitation, which was based on convenience sampling methods. Consequently, the majority of the study population were females, especially in the plant-based dietary groups, affecting the generalizability of our findings. Sex-differences has previously been reported as an influencing factor if simply evaluating absolute intake (g/d), and it has previously been suggested that energy-adjusted intakes should be evaluated as well [54]. To partially mitigate the concern of differences in energy reporting between females and males, dietary data in our study is expressed both as absolute intake (g/d) and as energy-adjusted intakes (g/MJ). Due power issues, test for differences split by sex within the plant-based dietary groups was not possible. Furthermore, similar to our study, previous studies have also reported that females are overrepresented in plant-based dietary groups [20, 55]. A higher proportion of females compared to males in plant-based groups has also been documented in a national report in Norway [19].

A limitation of the Norwegian version of the Myfood24 dietary assessment tool used in our study was the few available vegan/vegetarian products, which possibly impacted the reported intake of plant-based substitutes in the plant-based dietary groups. However, participants were able to report vegan/vegetarian foods in an open text option (described in the method, 24-hour dietary recall section), and energy intakes did not differ significantly between the groups. Although the lacto-ovo-vegetarians displayed the lowest energy intake. Another limitation was that six participants in the plant-based groups reported small amounts of animal-sourced foods in the 24-hour dietary recalls, inconsistent with the dietary practice based the electronic questionnaire and in the dietary screener (reported inclusion of animal source food the previous six months). This inconsistency could raise concerns about the accuracy of the dietary reporting. However, subjects who report small intake of animal-sourced foods not consistent with their plant-based diet has previously been reported [55, 56]. The study also took place during the COVID-19 pandemic, which could have impacted the recruitment and the eating habits of the participants, as previously discussed [24]. Finally, a limitation in the food group categorizations for the meat variables was that mixed dishes including meat (all types) were not included in the food subgroups red meat/white meat, however, the total weight of the dish reported by flexitarians and omnivores was reported in the manuscript. Consequently, an underestimation in the red and white meat variables is possible, if so, the underestimation is less than 50 g/d.

## Conclusion

Our study indicates that the participating 16-to-24-year-olds have risk of dietary shortcomings regardless of dietary practice. All dietary groups would benefit from increasing their consumption of vegetables, fruits and berries, and reducing their total intake from added and free sugar. The vegans reported the most favorable diet, characterized by the highest intake of vegetables, legumes, nuts and seeds, fruit and berries and vegetable oil. Consuming a variety of these food groups is emphasized to be important to achieve nutrient adequacy when following a plant-based diet, especially if omitting several animal-sourced foods. Additionally, vegans displayed the highest intake of dietary fiber, and the lowest consumption of saturated fatty acids supported by biomarkers. Furthermore, lacto-ovo-vegetarians had the lowest omega-3 intake supported by biomarkers. Our study findings should be confirmed in a larger study sample with youth in a Nordic population.

## Supporting information

**S1 Table. Food items included in the food groups in VeggiSkills-Norway.**
(DOCX)

**S2 Table. Changes made to food subcategories available in the Norwegian version of Myfood24.**
(DOCX)

**S3 Table. Food items included in the carotenoid-rich food categories in VeggiSkills-Norway.**
(DOCX)

**S4 Table. Median absolute food group intake among Norwegian youth with different dietary practice.**
(DOCX)

**S5 Table. Median energy-adjusted food group intake among Norwegian youth with different dietary practice.**
(DOCX)

**S6 Table. Mean absolute food group intake stratified by sex within dietary practice.**
(DOCX)

**S7 Table. Mean energy-adjusted food group intake in stratified by sex within dietary practice.**
(DOCX)

**S8 Table. Mean intake of carotenoid-rich foods among Norwegian youth with different dietary practice.**
(DOCX)

**S9 Table. Median intake of carotenoid-rich foods among Norwegian youth with different dietary practice.**
(DOCX)

## Acknowledgments

We thank Marte Haugen, Lale Marie Aasland, Stine Rambekk Henriksen, Silje Slettemoen, Christian Asbjørn Kvamsdal and Camilla Bjornes for being involved in the recruitment process.

## Author Contributions

**Conceptualization:** Synne Groufh-Jacobsen, Christel Larsson, Isabelle Mulkerrins, Dagfinn Aune, Anine Christine Medin.

**Formal analysis:** Synne Groufh-Jacobsen.

**Funding acquisition:** Christel Larsson, Anine Christine Medin.

**Investigation:** Synne Groufh-Jacobsen.

**Project administration:** Synne Groufh-Jacobsen.

**Supervision:** Christel Larsson, Anine Christine Medin.

**Writing – original draft:** Synne Groufh-Jacobsen.

**Writing – review & editing:** Synne Groufh-Jacobsen, Christel Larsson, Isabelle Mulkerrins, Dagfinn Aune, Anine Christine Medin.

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
