## [Decision Letter · Decision Letter 0]

4 Jun 2024

PONE-D-24-17355Food groups, macronutrient intake and objective measures of total carotenoids and fatty acids in 16-to-24-year-olds following different plant-based diets compared to an omnivorous dietPLOS ONE

Dear Dr. Groufh-Jacobsen,

Thank you for submitting your manuscript to PLOS ONE. After careful consideration, we feel that it has merit but does not fully meet PLOS ONE’s publication criteria as it currently stands. Therefore, we invite you to submit a revised version of the manuscript that addresses the points raised during the review process.

We look forward to receiving your revised manuscript.

Kind regards,

Rami Salim Najjar, Ph.D.

Academic Editor

PLOS ONE

Journal Requirements:

Reviewers' comments:

Reviewer's Responses to Questions

**Comments to the Author**

1. Is the manuscript technically sound, and do the data support the conclusions?

Reviewer #1: Yes

Reviewer #2: Partly

2. Has the statistical analysis been performed appropriately and rigorously? 

Reviewer #1: I Don't Know

Reviewer #2: Yes

3. Have the authors made all data underlying the findings in their manuscript fully available?

Reviewer #1: No

Reviewer #2: No

4. Is the manuscript presented in an intelligible fashion and written in standard English?

Reviewer #1: Yes

Reviewer #2: Yes

5. Review Comments to the Author

Reviewer #1: Title: Food groups, macronutrient intake and objective measures of total carotenoids and fatty acids in 16-to-24-year-olds following different plant-based diets compared to an omnivorous diet

This study reports on the dietary intake of youths (aged 16-24 years) consuming different dietary patterns and compares the intake based on food groups, macronutrients, and blood sample analysis of carotenoids and fatty acids.

Overall, the manuscript is well-written and organized.

Some points to address:

Methods:

Study Design and Study Population

• Can the authors provide a clear description of the timeline of the study? How much time elapsed between the food frequency questionnaire and the site visit with the first 24-hour dietary recall?

• Were blood samples collected at the same visit as the initial 24-hour dietary recall?

Classification into dietary practice

• Did participants also self-report their own dietary practice? How did it differ from their actual consumption?

Dietary Assessment

• Line 89: While the authors note that the dietary assessment has previously been described elsewhere, can they provide a brief description here?

• Were participants given instructions on how to use Myfood24 during the initial recall?

• Approximately how long does it take for an individual to go through the Myfood24?

Statistics

• Title line for Statistics should be moved to the next line.

• Line 159: “polysaturated” should be polyunsaturated

• Please include the value definitions of weak, moderate, and strong relationships regarding correlation coefficients.

Results:

• Table 2: Can the authors add separation between food groups to improve the visuals and a readability of the table? For example, group the plant-based foods, animal products, sweets, beverages, and alcohol.

• The authors report sex-differences as a limitation in the discussion. Could the authors further stratify their results based on sex for each dietary pattern? At a minimum, a descriptive analysis of the food groups and comparison to recommendations based on sex for energy intake would make the report more interesting.

Discussion:

• While the common definition of “youth” is 15-24 years, the average age in the report is 21 years (lowest was 20 and highest was 22). Much of the comparisons made are to studies with age ranges no higher than 18 years. This reviewer lacks knowledge on the cultural background of Norway – is there a large difference in lifestyle that could impact access to foods for those 16-18 versus those 19 and older? For example, those at university and living away versus those at home with family where food may be dictated by parental figures? Could these factors explain differences seen in these results versus those to which they are compared?

o Are there studies with more similar age ranges to which the authors can compare?

Supplemental Information

• Supplemental Table 2 : Ingredients is misspelled as “ingrediens”

Reviewer #2: This study has two primary objectives. The first objective is to investigate the impact of different diets on the food intake of Nordic youth and assess the extent to which these diets comply with dietary guidelines. The second objective is to validate self-reported fruit and vegetable consumption by comparing it with objective blood measurements of carotenoids. The strength of the study is its use of objective measurements of blood dietary markers. Also, the article is well-written and flows well. However, major issues include attempting to address two distinct research questions in a single paper, insufficient explanation and discussion regarding the studied population aged 16-24 years, and the methods used for classifying participants’ diets.

Major Issues:

1) Addressing participants' diet selection on their food consumption and macronutrient intake, while also attempting to validate a questionnaire for the objective assessment of fruit and vegetable intake, involves two distinct methodologies and topics that require substantial detail. Although this paper attempts to cover both, it lacks sufficient explanation in the introduction regarding the rationale for validating the self-report measure. Additionally, the methodology section does not clearly specify which carotenoids were examined or which food groups were expected to correlate with specific types of carotenoids. Furthermore, the results of the second research question are only presented in a supplementary table, adding to the confusion for readers trying to follow the discussion on these two separate research questions. Therefore, I strongly recommend that the authors remove the second research aim from this paper and consider expanding on it to submit as a separate publication.

2) This study included participants aged 16-24 years, but the rationale for selecting this age group is not provided in the introduction or methodology. This age range encompasses both adolescents and early adults. Based on the mean age presented in the results, it appears that most participants were college students (early adults). However, the discussion section predominantly focuses on children and adolescents, which does not accurately reflect the sample of this study. Therefore, I recommend adding a detailed explanation in the introduction regarding the selection of the sample and improving the discussion by incorporating studies conducted on college students and late adolescents.

3) The biggest concern I have regarding the findings is the method used for classifying participants' diets (Lines 75-79). Researchers determined the participants' diets based on their responses to the food frequency questionnaire. However, this classification was not confirmed with the participants (or if it was, it was not clearly stated in the paper). Consequently, six participants were later identified as not adhering to their diet, which may not be due to inconsistency in their diet but rather incorrect initial classification. The authors need to justify that their classification method accurately represents the participants’ actual diets.

Minor Issues:

4) In the abstract (Line 9), provide what blood markers collected. Example: provided dried blood samples (DBS) to analyze blood fatty acids and carotenoids.

5) Line 20-21: The authors did not examine the causality between nutrition education and dietary intake. Therefore, they cannot assert that education might benefit participants to improve their diet. Instead, they should concentrate on identifying which diet group demonstrated superior outcomes in terms of healthy food intake and macronutrient consumption compared to others.

6) In lines 46-47, the authors mention that the association between diet and food intake has already been studied, and in the following paragraph, they highlight the innovation of the current study, particularly in its focus on a Nordic youth sample. However, the rationale for why the results might differ in a Nordic youth sample is missing.

7) Again, there is no background regarding the research aim provided in line 56-57. I recommend authors to remove this aim.

8) In line 81-84: What happened to those participants? Did you remove them from the analysis? Or how did you address their inconsistent diet?

9) In line 90, the paper provided for further details is under review. I can not look at the details of the method utilized for dietary assessment. If the cited paper is not published, the authors should provide details in this paper.

10) Line 97-99: This should be presented in the results, not in the methods.

11) Line 109-110: Did participants complete the 24-dietary recall by themselves or over a phone interview? Please provide more details regarding the remaining 3 days of data collection.

12) Line 111-112: How many weekdays and weekends have each participant completed?

13) Line 113-118: I have a general concern regarding the selection of food groups, particularly the inclusion of the meat food group. This inclusion is problematic since participants categorized as vegan, lacto-ovo vegetarians, and pescatarians should not consume any meats. Therefore, there is no hypothesis associated with this food group, as the definition of the independent variable requires the dependent variable to be null. The results presented in Table 2 support this observation, as meat intake is either 0 or very low for the aforementioned groups, while it is higher in the other groups. I recommend that the authors provide a justification for their selection of food groups and consider including only foods that can be consumed in plant-based diets and omnivore diets. This adjustment will not only clarify the hypothesis but also reduce the number of models run, thereby increasing the power of the study.

14) Line 131: What types of carotenoids were analyzed?

15) Line 138: What types of fatty acids were analyzed?

16) In the statistics section, there is no justification or description of the sample size selection. While I understand that this is a secondary study and the original study may not have been powered specifically for this question, it's important to provide some insight into the extent to which this study has power for the proposed analysis with the included sample size for the cross-sectional study. I recommend including a statement addressing this aspect.

17) Line 247: The results are presented for each carotenoid level, yet the introduction does not provide a rationale for why and which types of carotenoids are related to youth dietary intake. If these results are to be presented, the authors should introduce the types of carotenoids and their relationship with child dietary intake.

18) As previously mentioned, the discussion would benefit from incorporating findings from younger adults, given the sample demographics of the current study.

19) Line 414: Please revised as Finnish adults

20) Line 489-491: This study cannot draw such conclusions as there was no educational nutrition intervention. Moving forward, the authors should focus on offering recommendations regarding which types of diets show promise for promoting healthy dietary intake or what strategies they recommend to youth in different diet groups to enhance their dietary habits.

6. PLOS authors have the option to publish the peer review history of their article (what does this mean?). If published, this will include your full peer review and any attached files.

Reviewer #1: No

Reviewer #2: **Yes: **Aliye Cepni

---

## [Author Response · Author response to Decision Letter 0]

22 Aug 2024

The revised manuscript, along with a detailed point-by-point response to each comment, has been uploaded within the specified timeline. I believe the revisions have significantly improved the quality of the paper and hope it now meets the journal's standards. Thank you for the reviewers' constructive feedback.

Please see reviewer response in the uploaded point-by-point letter.

---

## [Decision Letter · Decision Letter 1]

13 Sep 2024

Food groups, macronutrient intake and objective measures of total carotenoids and fatty acids in 16-to-24-year-olds following different plant-based diets compared to an omnivorous diet

PONE-D-24-17355R1

Dear Dr. Groufh-Jacobsen,

We’re pleased to inform you that your manuscript has been judged scientifically suitable for publication and will be formally accepted for publication once it meets all outstanding technical requirements.

Kind regards,

Rami Salim Najjar, Ph.D.

Academic Editor

PLOS ONE

Additional Editor Comments (optional):

Reviewers' comments:

Reviewer's Responses to Questions

**Comments to the Author**

1. If the authors have adequately addressed your comments raised in a previous round of review and you feel that this manuscript is now acceptable for publication, you may indicate that here to bypass the “Comments to the Author” section, enter your conflict of interest statement in the “Confidential to Editor” section, and submit your "Accept" recommendation.

Reviewer #1: All comments have been addressed

Reviewer #2: All comments have been addressed

2. Is the manuscript technically sound, and do the data support the conclusions?

Reviewer #1: Yes

Reviewer #2: Yes

3. Has the statistical analysis been performed appropriately and rigorously? 

Reviewer #1: Yes

Reviewer #2: Yes

4. Have the authors made all data underlying the findings in their manuscript fully available?

Reviewer #1: Yes

Reviewer #2: No

5. Is the manuscript presented in an intelligible fashion and written in standard English?

Reviewer #1: Yes

Reviewer #2: (No Response)

6. Review Comments to the Author

Reviewer #1: The authors clearly addressed the comments and improved the overall structure of the manuscript. It is well-written and ready for publication.

Reviewer #2: Thank you for thoroughly addressing the suggestions provided in the initial review. I appreciate the thoughtful revisions, which have fully met my expectations.

Although this does not affect my overall decision, I recommend that the authors make the minor edits noted before proceeding with publication.

Minor:

Introduction 58-59: Suggest editing it as “The United Nations (UN) defines youth as individuals between the ages of 15 and 24, marking a transitional phase from the dependence of childhood to the independence of adulthood (cite).”

Introduction 53-62: The authors should justify the need for studying the Nordic population. It could be through highlighting the distinct cultural differences in dietary practices compared to other countries where similar relationships have already been explored. By emphasizing the unique characteristics of the Nordic diet and lifestyle.

7. PLOS authors have the option to publish the peer review history of their article (what does this mean?). If published, this will include your full peer review and any attached files.

Reviewer #1: No

Reviewer #2: No

---

## [Editor Report · Acceptance letter]

17 Sep 2024

PONE-D-24-17355R1 

PLOS ONE

Dear Dr. Groufh-Jacobsen, 

I'm pleased to inform you that your manuscript has been deemed suitable for publication in PLOS ONE. Congratulations! Your manuscript is now being handed over to our production team.

Kind regards, 

on behalf of

Dr. Rami Salim Najjar 

Academic Editor

PLOS ONE